# Quantized Alternate Current on Curved Graphene

**Kyriakos Flouris [1,*], Sauro Succi [2,3] and Hans J. Herrmann [4,5]**

[1]   ETH Zürich, Computational Physics for Engineering Materials, Institute for Building Materials,
      Wolfgang-Pauli-Str. 27, HIT, CH-8093 Zürich, Switzerland
[2]   Center for Life Nano Sciences at La Sapienza, Istituto Italiano di Tecnologia, viale R. Margherita, 265,
      00161 Roma, Italy; sauro.succi@gmail.com
[3]   Institute for Applied Computational Science, J. Paulson School of Engineering and Applied Sciences,
      Harvard University, 29 Oxford Street, Cambridge, MA 02138, USA
[4]   Departamento de Física, Universidade do Ceará, 60451-970 Fortaleza, Brazil; hans@ifb.baug.ethz.ch
[5]   Centre national de la recherche scientifique, UMR 7636, PMMH, ESPCI, 10 rue Vauquelin,
      75231 Paris CEDEX 05, France
*    Correspondence: kyriakos.flouris@cantab.net

**Abstract:** Based on the numerical solution of the Quantum Lattice Boltzmann Method in curved space, we predicted the onset of a quantized alternating current on curved graphene sheets. This numerical prediction was verified analytically via a set of semi-classical equations that related the Berry curvature to real space curvature. The proposed quantized oscillating current on curved graphene could form the basis for the implementation of quantum information-processing algorithms.

**Keywords:** graphene; electronic transport; quantum cellular automata; curved space; quantum lattice Boltzmann; Dirac fermion; Bloch oscillations; Berry curvature

## 1. Introduction

In recent years, the most puzzling features of quantum mechanics that have long been regarded as sort of extravagant speculations, such as entanglement and non-local "spooky action at a distance", have received spectacular experimental confirmation [1–4]. Besides their deep fundamental implications, such phenomena may also open up transformative scenarios for material science and related applications in quantum computing and telecommunications [5,6]. Along with such bursts of experimental activity, a corresponding upsurge of theoretical and computational methods has also emerged in the last two decades, including, among others, new quantum many-body techniques, quantum simulators [7–10], and quantum walks [11].

Quantum walks were first introduced by Aharonov and collaborators in 1993 [12], just a few months before the appearance of the first Quantum Lattice Boltzmann scheme [13], which was only recently recognized to be a quantum walk [14]. Quantum walks [15,16] are currently utilized to investigate exotic states of quantum matter [17] and to design new materials and technologies for quantum engineering applications [18].

Quantum walks can also help explore the emergence of classical behavior in the limit of a vanishing de Broglie length [19]. Likewise, quantum cellular automata [20–22], can be used for simulating complex systems in analogue with their classical counterparts.

Finally, quantum walks have also shown connections with topological aspects of quantum mechanics, most notably the Berry phase [23]. Indeed, Berry connection and Berry curvature can be understood as a local gauge potential and gauge field, respectively, and they define a Berry phase as introduced in 1984 [23]. The Berry phase has important implications as an analytical tool in topological phases of matter [24] and, under suitable conditions, it can also be related to real space curvature [25], thus providing a potential bridge between the classical and quantum descriptions of a given system.

As of quantum materials, graphene presents one of the most promising cases for realizing a new generation of quantum devices [26–28]. Indeed, since its discovery [29], this flatland wonder material has not ceased to surprise scientists with its amazing mechanical and electronic behavior. For example, stacking graphene sheets at specific angles has shown spectacular indications of superconductivity and other exotic properties [30].

Tunable transport properties are a basic requirement in electronic devices and specifically in graphene [31]. Furthermore, it has been shown that graphene sheets can be curved in such a way as to trap particles [32], thus opening further prospects for technological applications based on localized quantum states.

In this work, we propose the generation of a quantized oscillating current on curved graphene, which could be used in conjunction with trapped fermions for the realization of quantum cellular automata.

Electron transport is simulated by numerically solving the Dirac equation in curved space [32,33] using an extension to curved space of the Quantum Lattice Boltzmann Method [13]. In addition, a simpler representation of the system is solved analytically through a set of semi-classical equations of motion, relating Berry to real space curvature.

The paper is organized as follows. First, we introduce the Dirac equation and its extension to curved space and specifically to deformed graphene. In the subsequent section, we present the results of numerical simulations. Finally, we conclude with a summary and outlook section. A detailed description of the numerical model is provided in the Appendix (Appendix A).

## 2. The Dirac Equation in Curved Space and Graphene

The Dirac equation in curved space can be written in compact notation as follows:

$$(i\gamma^\mu D_\mu - m)\Psi = 0, \tag{1}$$

in natural units $\hbar = c = 1$, where $m$ is the particle rest mass, and the index $\mu = 0, 1, 2$ runs over 2D space-time. In the above, $\Psi = (\Psi^+, \Psi^-) = (\psi_1^+, \psi_2^-, \psi_1^-, \psi_2^+) \in \mathbb{C}^4$ denotes the Dirac four-spinor, and $\gamma^\mu = \gamma^\alpha e_\alpha^{\ \mu}$ are the generalized $\gamma$-matrices, where $\gamma^\alpha \in \mathbb{C}^{4\times4}$ are the standard $\gamma$-matrices (in Dirac representation). The symbol $e_\alpha^{\ \mu}$ is the tetrad (first index: Flat Minkowski, second index: Curved space-time).

Here, the tetrad is defined by $e_\alpha^\mu g_{\mu\nu} e_\beta^\nu = \eta_{\alpha\beta}$ [34], where $g_{\mu\nu}$ denotes the metric tensor and $\eta_{\alpha\beta}$ is the Minkowski metric. The tetrad basis is chosen such that the standard Dirac matrices can be utilized with no need to transform to a new coordinate basis. The symbol $D_\mu$ denotes the covariant spinor derivative, defined as $D_\mu \Psi = \partial_\mu \Psi + \Gamma_\mu \Psi$, where $\Gamma_\mu$ denotes the spin connection matrices given by $\Gamma_\mu = -\frac{i}{4}\omega_\mu^{\alpha\beta}\sigma_{\alpha\beta}$, where $\sigma_{\alpha\beta} = \frac{i}{2}[\gamma_\alpha, \gamma_\beta]$ and $\omega_\mu^{\alpha\beta} = e_\nu^\alpha \nabla_\mu e^{\nu\beta}$. The Dirac equation in curved space describes quantum relativistic Dirac particles (e.g., electrons ) moving on arbitrary manifold trajectories.

The covariant derivative ensures the independence of the Dirac equation of the coordinate basis. The covariance is satisfied by the connection coefficients, which can be interpreted physically as a vector potential. The Poincare symmetries are obeyed by the Dirac equation, ensuring the special relativistic nature of the wavefunctions. The mass term represents the Minkowski metric invariant rest mass.

Interactions add to an effective mass by the very definition of a covariant derivative, which places the vector potential on the same mathematical basis as a physical mass. Graphene is modeled by a massless Dirac Hamiltonian.

*Theory of Strained Graphene*

Using the tight-binding Hamiltonian to describe the bi-partite lattice of graphene, it is established that in the low-energy limit, the dispersion relation is linear, as described by the Dirac cones at the corners of the first Brillouin zone, which can be described by the following Dirac Hamiltonian:

$$H_D = -i \int \Psi^\dagger \gamma^0 \gamma^i \partial_i \Psi d^2 x, \tag{2}$$

in natural units, where $\Psi$ is in the chiral representation. In the context of graphene, the general Dirac spinor is defined as $\Psi = (\Psi_a^K, \Psi_a^{K'}) = (\psi_A^K, \psi_B^{K'}, \psi_A^{K'}, \psi_B^K)$, for sub-lattices $A, B$ and valleys $K, K'$. The equation of motion stemming from this Hamiltonian is precisely the Dirac equation.

In this work, we consider a static space-time metric, with trivial time components:

$$g_{\mu\nu} = \begin{pmatrix} 1 & 0 \\ 0 & -g_{ij} \end{pmatrix},$$

where the latin indices run over the spatial dimensions. This simplifies the Dirac Equation (1) to:

$$\partial_t \Psi + \sigma^a e_a{}^i (\partial_i + \Gamma_i)\Psi = 0 - i\gamma^0 m \Psi, \tag{3}$$

with $\sigma^a = \gamma^0 \gamma^a$. After the addition of external vector and scalar potentials $A_i(x)$ and $V(x)$ respectively, as explained in Reference [35], the Dirac equation takes the following form:

$$\partial_t \Psi + \sigma^a e_a{}^i (\partial_i + \Gamma_i - iA_i)\Psi = -i\gamma^0 (m - V)\Psi. \tag{4}$$

Defining the Dirac current as $J^\mu = \overline{\Psi}\gamma^\mu \Psi$, the charge density conservation law can be written as $\partial_t \rho + \nabla_i J^i = 0$, where $\rho = \Psi^\dagger \Psi \in \mathbb{R}$ and $J^i = \overline{\Psi}\gamma^i \Psi \in \mathbb{R}$.

The standard Dirac Hamiltonian for Equation (4) is given by:

$$H_D = -i \int \Psi^\dagger \sigma^a e_a{}^i (\partial_i + \Gamma_i - iA_i)\Psi \sqrt{g} d^2 x. \tag{5}$$

For the case of graphene, the effective Hamiltonian reads as follows [36]:

$$H_D^* = -i \int \Psi^\dagger \sigma^a (v_a^{*i}\partial_i + \Gamma_a^* - iA_a^*)\Psi d^2 x, \tag{6}$$

where $v_a^{*i} = \delta_{ai} + u_{ai} - \beta\varepsilon_{ai}$ is the space-dependent Fermi velocity, $\Gamma_a^* = \frac{1}{2}\partial_j v_a^{*j}$ is a complex gauge vector field that guarantees the hermicity of the Hamiltonian, and $A_a^*$ is a strain-induced pseudo-vector potential, given by $A_a^* = (A_x^*, A_y^*) = \frac{\beta}{2a}(\varepsilon_{xx} - \varepsilon_{yy}, -2\varepsilon_{xy})$. Furthermore, $\beta$ is the material-dependent electron Grueneisen parameter, $a$ is the lattice spacing, and $\varepsilon_{ij} = u_{ij} + \frac{1}{2}\partial_i h \partial_j h$ is the general strain tensor, with in-plane, $u_{ij}$ and out-of-plane, $h$ deformations.

Comparing this to the standard Dirac Hamiltonian in curved space Equation (5), we can match both Hamiltonians $H_D$ and $H_D^*$ by fulfilling the following relations:

$$v_a^{*i} = \sqrt{g}e_a{}^i, \quad \Gamma_a^* = \sqrt{g}e_a{}^i\Gamma_i, \quad A_a^* = \sqrt{g}e_a{}^i A_i. \tag{7}$$

All three relations above can be simultaneously fulfilled by an effective metric tensor derived from the explicit expression of the tetrad [35]. The numerical solutions are obtained with the Quantum Lattice Boltzmann Method, as described in Appendix A and Reference [35].

## 3. Quantized Alternating Current Graphene Strip

To investigate the potential of curvature on curved graphene sheets, we propose a periodic system with alternating current (AC) behavior, which is quantized according to its shape. The system geometry is initialized by the discrete mapping (or chart),

$$
h^\alpha(x,y) = \begin{pmatrix} x \\ y \\ y\sin(\eta x/2) \end{pmatrix}
\tag{8}
$$

with $x \in \{0, 2\pi\}$, $y \in \{-L_y/2, L_y/2\}$, $L_y$ being the domain size in the $y$ dimension, see Figure 1. The boundaries are periodic along the $x$-direction and closed at $-L_y/2$, $L_y/2$.

The initial condition is given by a Gaussian wavepacket of the form:

$$
\Psi(\mathbf{r}, \mathbf{k}) = \frac{1}{\sqrt{2\pi\zeta^2}} \begin{pmatrix} 1 \\ \lambda e^{i\theta} \end{pmatrix} e^{-\frac{|\mathbf{r}|^2}{4\zeta^2} + i\mathbf{k}\cdot\mathbf{r}},
\tag{9}
$$

where $\lambda = \pm 1$ is the band index, $\theta = \arctan(k_y/k_x)$, $\zeta$ is a measure of the width, $\mathbf{r} = (x,y)$, $x$, $y$ are the two space coordinates, and $\mathbf{k} = (k_x, k_y)$, $k_x$, $k_y$ represent the $x$ and $y$ momenta, respectively. The initial values are taken as $k_x = 1$, $k_y = 0$, and $\lambda = 1$. In the simulations, we consider a rectangular sheet with periodic boundary conditions on a grid size of $L_x \times L_y = 256 \times 128$ or 20 nm $\times$ 5 nm, while the external potential $A_a$ is set to zero. Therefore, the subsequent motion is purely curvature driven.

The discretization of the real space shape of the graphene strip is plotted in Figure 1a. The norm of the wavefunction, $\|\Psi\|^2$, i.e., the probability density, $\rho$ is plotted in Figure 1b for the initial and few subsequent time-steps. As one can appreciate, the wavepacket spreads as expected, with no clear indication of motion along the $y$ direction.

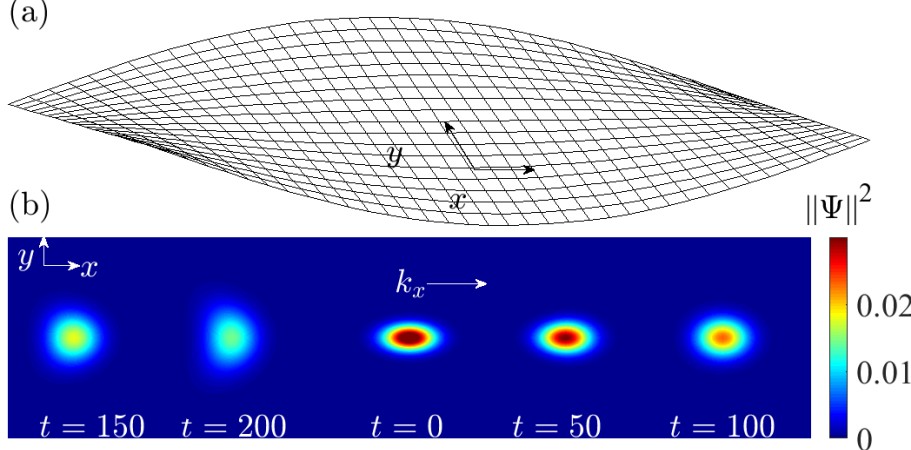

**Figure 1.** (**a**) Real space geometry of the graphene strip, $x, y$ denote the coordinate directions. (**b**) Density plots of the wavepacket for different time-steps, the $k_x$ arrow denotes the propagation direction. The bulk of the wavepacket propagates forward and is spreading as expected.

The position of the center of charge density along the $y$ direction:

$$\bar{y} = \left( \int^{area} y(\rho(t) - \rho(0))dA \right) / \int^{area} \rho(t)dA, \tag{10}$$

is plotted as a function of time in Figure 2, where $dA = dxdy$. A small but significant oscillation along the $y$ direction is observed.

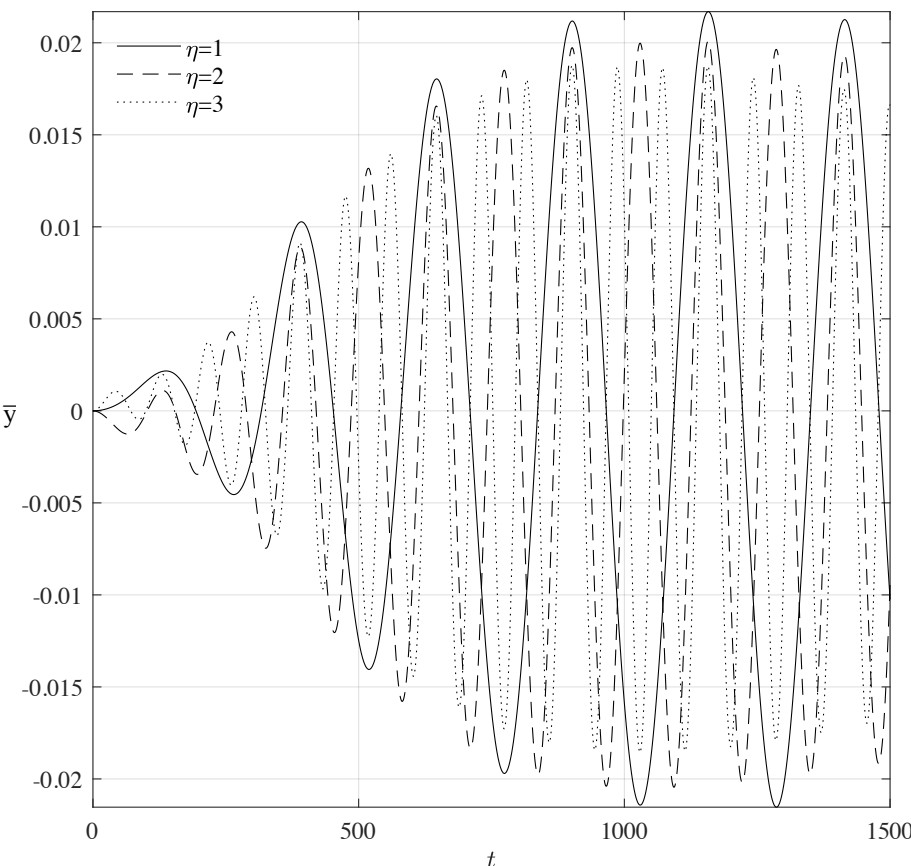

**Figure 2.** Time evolution of relative position of the center of charge in the radial direction for geometries with $\eta = 1, 2, 3$.

These oscillations can be understood as the geometrical equivalent of the Bloch oscillations and they are a consequence of the sinusoidal, periodic domain, with the frequency quantized in units of the parameter $\eta$. For a slowly perturbed Hamiltonian, expanding around the wavepacket center $c$ (initialized to $(0,0)$ here), $H = H_c + \Delta H$, assuming a periodic system described by a Bloch wavefunction, the semi-classical equations of motion are given by [37]:

$$\dot{\mathbf{r}}_c = \frac{\partial \varepsilon}{\partial \mathbf{k}} - (\Omega_{\mathbf{kr}} \cdot \dot{\mathbf{r}}_c + \Omega_{\mathbf{kk}} \cdot \dot{\mathbf{k}}_c) - \Omega_{\mathbf{k}t} \tag{11}$$

$$\dot{\mathbf{k}}_c = \frac{\partial \varepsilon}{\partial \mathbf{r}} - (\Omega_{\mathbf{rk}} \cdot \dot{\mathbf{k}}_c + \Omega_{\mathbf{rr}} \cdot \dot{\mathbf{r}}_c) - \Omega_{\mathbf{r}t} \tag{12}$$

where $\mathbf{r}_c, \mathbf{k}_c$ are the center of mass position and momentum of the wavepacket, $\mathbf{r}, \mathbf{k}$ are the position and momentum vectors, $t$ is time, $\varepsilon$ is the band energy, and $\Omega_{\mathbf{kr}} = (\Omega_{\mathbf{kr}})_{\alpha\beta} = \partial \mathbf{k}_\alpha \mathcal{A}_{\mathbf{r}_\beta} - \partial \mathbf{r}_\beta \mathcal{A}_{\mathbf{k}_\alpha}$ is the Berry curvature and $\mathcal{A}_{\mathbf{r}}$ the Berry connection.

As shown in Appendix B, the Berry phase, and thus Berry curvature, can be related directly to the spin connection $\Gamma_\mu$ through $\mathcal{A}_n^i(\mathbf{R}) = i\langle\Psi(\mathbf{R})|\partial_{\mathbf{R}}|\Psi(\mathbf{R})\rangle \implies \mathcal{A}_{R_i} = \mathrm{Tr}\langle\Psi|\Gamma_i|\Psi\rangle$ for some parameter space $\mathbf{R}$ and eigen-function index, $n$.

The non-zero terms of Equations (11) and (12) for the specific geometry are $\dot{\mathbf{r}}_c = \partial\varepsilon/\partial\mathbf{k} \approx v_f$, $\dot{\mathbf{k}}_c = \Omega_{\mathbf{rr}} \cdot \dot{\mathbf{r}}_c$, which imply:

$$\frac{\partial \mathbf{k}_\alpha}{\partial \mathbf{r}_\beta} = (\Omega_{\mathbf{rr}})_{\alpha\beta}. \tag{13}$$

For small-amplitude local wavepackets:

$$\delta\mathbf{k}_\alpha = \int (\Omega_{\mathbf{rr}})_{\alpha\beta} dr_\beta = \int (\partial\mathbf{r}_\alpha \mathcal{A}_{\mathbf{r}_\beta} - \partial\mathbf{r}_\beta \mathcal{A}_{\mathbf{r}_\alpha}) d\mathbf{r}_\beta \tag{14}$$

and thus, $\delta\mathbf{k}_y \propto sin(\eta x)$ and $\delta\mathbf{k}_x \propto cos(\eta x)$. Therefore, the oscillations can be explained in terms of a real space Berry curvature, jointly with the classical geodesic equation on the corresponding manifold.

The frequency of these Bloch-like oscillations is quantized according to $\eta$. Finally, some forward moving charge, even if driven, will experience an equivalent transverse oscillating motion, therefore the system might be implemented as a periodic, quantized oscillating current device.

## 4. Conclusions and Outlook

We have proposed the realization of a quantized alternating current on a curved graphene sheet. The oscillating current was numerically computed through the Quantum Lattice Boltzmann Method in curved space and verified analytically via a set of semi-classical equations, relating the Berry curvature to real space curvature. We interpreted this result as a geometrical analogue of the Bloch oscillations, quantized according to the geometrical period $\eta$.

Building on these results, more complex and adjustable graphene devices can be envisaged in the context of curvature-based design. For example, the proposed quantized oscillating current on graphene, in conjunction with trapping quantum dots [32], could form the building block for quantum information-processing algorithms.

**Author Contributions:** Conceptualization, K.F.; methodology, K.F.; software, K.F.; validation, K.F. and S.S.; formal analysis, K.F.; investigation, K.F.; resources, K.F. and H.J.H.; data curation, K.F.; writing–original draft preparation, K.F.; writing–review and editing, K.F., S.S. and H.J.H.; visualization, K.F.; supervision, S.S and H.J.H.; project administration, S.S. and H.J.H.; funding acquisition, H.J.H.

**Funding:** K.F. and H.J.H. are grateful for the financial support of the Swiss National Science Foundation, under Grant No. 200021 165497 and to FUNCAP and CAPES for their support. S.S. acknowledges funding from the European Research Council under the European Union Horizon 2020 Framework Programme (No. FP 2014-2020) ERC Grant Agreement No. 739964 (COPMAT).

**Acknowledgments:** K.F. thanks Ioannis Petrides for helpfull conversations.

**Conflicts of Interest:** The authors declare no conflict of interest.

## Appendix A. Curved-Space Quantum Lattice Boltzmann

The Quantum Lattice Bolzmann (QLB) Method used for solving the Dirac equation as minimally coupled to curved space is an extension of the original method developed by Succi et al. [13]. The method exploits the conceptual similarities between the Dirac equation and the Boltzmann equation on the lattice.

We present here the QLB method for a three-dimensional manifold, with straight forward usage to lower dimensional systems, [38–40].

*Appendix A.1. The Dirac Equation*

The classical Boltzmann equation for a particle density distribution function $f(x_a, v_a, t)$ is given by:

$$\partial_t f + v^i \partial_{x^i} f = \mathcal{C}[f] - F^a \partial_{v^a} f, \tag{A1}$$

the left-hand side describes the advection of the distribution function, velocity $v^a$, whereas the right-hand side describes the collisions between particles and the effect of external forces $F^a$. Furthermore, the Dirac equation in curved space in Equation (1) can be cast into a kinetic theory form,

$$\partial_t \Psi + \sigma^a \partial_a \Psi = \mathcal{C}\Psi + \mathcal{F}\psi. \tag{A2}$$

Therefore, similarly to the Boltzmann equation, the left-hand side represents the 'free streaming' step along matrix valued 'velocities' $\sigma^i$ while the right-hand side contains a 'collision' and a 'forcing' term.

The collision term of Equation (A2) is represented by:

$$\mathcal{C} = -(im\gamma^0 + \sigma^a e_a^i \Gamma_i), \tag{A3}$$

where $m$ is the fermion mass. The 'forcing term' is given by:

$$\mathcal{F} = -\sigma^a (e_a^i - \delta_a^i)\partial_i. \tag{A4}$$

where the symbols have their usual meaning. The partial derivative of the Dirac equation is distributed between the streaming part and the forcing term, resulting in a lattice-compatible classical streaming operator of the form $\partial_t + v^a \partial_a$, where $v^a \in \mathbb{Z}$. The forcing term is a consequence of the generalized Dirac matrices $\gamma^i = e_a^i \gamma^a$ and captures the bulk of the curvature effects. The partial derivative in Equation (A4) is approximated by a local lattice finite difference scheme .

*Appendix A.2. Diagonal Streaming Operator*

In order to obtain a diagonal streaming operator, the complex $\sigma$-matrices have to be diagonalized first, which yields a diagonal velocity matrix with eigenvalues $v^a = \pm 1$. The diagonalization is achieved by suitable "rotation matrices":

$$X_a^\dagger \sigma^a X_a = \begin{pmatrix} 1 & 0 & 0 & 0 \\ 0 & 1 & 0 & 0 \\ 0 & 0 & -1 & 0 \\ 0 & 0 & 0 & -1 \end{pmatrix} = \gamma^0 \qquad \text{for } a = 0, 1, 2,$$

where the unitary transformation matrices $X_1, X_2, X_3$ are given by:

$$X_1 = \frac{1}{\sqrt{2}} \begin{pmatrix} 1 & 0 & -1 & 0 \\ 0 & 1 & 0 & -1 \\ 0 & 1 & 0 & 1 \\ 1 & 0 & 1 & 0 \end{pmatrix}, \quad X_2 = \frac{1}{\sqrt{2}} \begin{pmatrix} 0 & i & 0 & 1 \\ -i & 0 & i & 0 \\ -1 & 0 & -1 & 0 \\ 0 & -1 & 0 & -i \end{pmatrix},$$

$$X_3 = \frac{1}{\sqrt{2}} \begin{pmatrix} 1 & 0 & 0 & -1 \\ 0 & 1 & 1 & 0 \\ 1 & 0 & 0 & -1 \\ 0 & 1 & 1 & 0 \end{pmatrix}.$$

The streaming and collision operations are performed in successive steps using operator splitting, since the simultaneous diagonalization of the three $\sigma$ matrices is not possible:

$$\Psi\left(t + \frac{\Delta t}{D}\right) = \exp\left(-\Delta t \sigma^1 \partial_1 + \frac{\Delta t}{D}(\mathcal{C} + \mathcal{F})\right)\Psi(t),$$

$$\Psi\left(t + \frac{2\Delta t}{D}\right) = \exp\left(-\Delta t \sigma^2 \partial_2 + \frac{\Delta t}{D}(\mathcal{C} + \mathcal{F})\right)\Psi\left(t + \frac{\Delta t}{D}\right),$$

$$\Psi(t + \Delta t) = \exp\left(-\Delta t \sigma^3 \partial_3 + \frac{\Delta t}{D}(\mathcal{C} + \mathcal{F})\right)\Psi\left(t + \frac{2\Delta t}{D}\right),$$

where $D = 3$ denotes the spatial dimensions. Each streaming step can be diagonalized by left multiplying with $X_a^\dagger$.

$$X_a^\dagger \Psi\left(t + \frac{\Delta t}{D}\right) = \exp\left(-\Delta t \sigma^a \partial_a + \Delta t(\tilde{\mathcal{C}}_a + \tilde{\mathcal{F}}_a)\right)\check{\Psi}_a(t), \tag{A5}$$

with the definitions:

$$\tilde{\Psi}_a := X_a^\dagger \Psi, \quad \tilde{\mathcal{F}}_a := \frac{1}{2} X_a^\dagger \mathcal{F} X_a, \quad \tilde{\mathcal{C}}_a := \frac{1}{2} X_a^\dagger \mathcal{C} X_a,$$

for $a = 1, 2, 3$ (no Einstein summation is used here). The exponential approximated as:

$$\exp\left(-\Delta t \sigma^a \partial_a + \Delta t(\tilde{\mathcal{C}} + \tilde{\mathcal{F}})\right)$$
$$\approx \left(\mathbb{I} - \Delta t \sigma^a \partial_a + \Delta t(\tilde{\mathcal{C}}_a + \Delta t \tilde{\mathcal{F}}_a)\right)$$
$$+ \left(\mathbb{I} - \frac{\Delta t}{2}\tilde{\mathcal{C}}_a\right)^{-1}\left(\mathbb{I} + \frac{\Delta t}{2}\tilde{\mathcal{C}}_a\right)\right)$$

The expansion of the collision operator $e^{\Delta t \tilde{\mathcal{C}}_a}$ is unitary and thus conserves exactly the probability of the wavefunction. The streaming $e^{-\Delta t \gamma^0 \partial_a}$ and forcing $e^{\Delta t \tilde{\mathcal{F}}_a}$ operators are not expanded, as this is prohibited by the derivative. A simple $2^{nd}$-order expansion is performed, limiting the probability norm to $\Delta t^2$ accuracy. The operator splitting implies an error of order $\mathcal{O}(\Delta t^2)$, as $e^{\Delta t X} \cdot e^{\Delta t Y} = e^{\Delta t(X+Y) + 1/2 \Delta t^2 [X,Y]} = e^{\Delta t(X+Y)} + \mathcal{O}(\Delta t^2)$.

The manifold is described by a chart $h$ defined in linear space, discretized on a regular rectangular lattice. The curved space Quantum Lattice Boltzmann Method evolves the four-spinor $\Psi = (\Psi^+, \Psi^-) = (\Psi_1^+, \Psi_2^-, \Psi_1^-, \Psi_2^+)$ from $t$ to $t + \delta t$. Once the operators are split, the following algorithm is performed in sequence for each lattice direction $n_a$, where $n_1 = (1, 0)$, $n_2 = (0, 1)$, and $a = 1, 2$.

1. **Rotation:** The spinor is rotated by $X_a$,

$$\tilde{\Psi}_a(x,t) = X_a^\dagger \Psi(x,t). \tag{A6}$$

2. **Collisions and curvature:** The collision and force operators are applied to the rotated spinor,

$$\tilde{\Psi}_a^*(x,t) = \left(\Delta t \tilde{\mathcal{F}}_a + (\mathbb{I} - \frac{\Delta t}{2}\tilde{\mathcal{C}}_a)^{-1}(\mathbb{I} + \frac{\Delta t}{2}\tilde{\mathcal{C}}_a)\right)\tilde{\Psi}_a(x,t),$$

where $\tilde{\Psi}_a^*(x,t)$ denotes an auxiliary field,

$$\tilde{\mathcal{C}}_a = \frac{1}{2}X_a^\dagger \mathcal{C} X_a = -\frac{i}{D}m(X_a^\dagger \gamma^0 X_a) - \gamma^0 e_a^i \Gamma_i, \tag{A7}$$

$$\tilde{\mathcal{F}}_a \tilde{\Psi}_a(x,t) = \left(e_a^i - \delta_a^i\right)\left(\tilde{\Psi}_a(x \mp n_i \Delta t, t) - \tilde{\Psi}_a(x,t)\right), \tag{A8}$$

where $n_i$ is the lattice direction and $\mathcal{C}$ is the collision term, Equation (A3). The upper sign applies to the spin-up components $(\Psi_1^+, \Psi_2^+)$ and the lower sign to the spin-down components $(\Psi_1^-, \Psi_2^-)$.

3. **Streaming:** The spinor components are streamed to the closest grid points along the lattice direction $\pm n_a$,

$$\tilde{\Psi}_a(x, t + \frac{\Delta t}{2}) = \tilde{\Psi}_a^*(x \mp n_a \Delta t, t). \tag{A9}$$

4. **Inverse Rotation:** The spinor is rotated back via $X_a$,

$$\Psi_a(x, t + \frac{\Delta t}{2}) = X_a \tilde{\Psi}_a(x, t + \frac{\Delta t}{2}). \tag{A10}$$

5. Repeat steps 2–4 for the next spatial direction.

The external potentials $V(x)$, scalar, and $A(x)$, vector are added to the collision operator Equation (A7), such that:

$$\tilde{\mathcal{C}}_a = \frac{1}{2}X_a^\dagger \mathcal{C} X_a = -\frac{i}{D}(m - V)(X_a^\dagger \gamma^0 X_a) - \gamma^0 e_a^i(\Gamma_i - iA_i). \tag{A11}$$

The simulation for strained graphene is carried out with modified Equations (A7) and (A8), according to the following scheme:

$$\tilde{\mathcal{C}}_a \to \sqrt{g}\tilde{\mathcal{C}}_a, \; e_a^i \to \sqrt{g}e_a^i.$$

The additional factor $\sqrt{g}$ originates from the volume element of the Hamiltonian Equation (6).

## Appendix B. Berry Phase Relation to the Spin Connection

To solve the Dirac equation, minimally coupled to curvature, Equation (1), with $A_i = 0$ and assuming that the wavepacket has a negligible spread, $\delta \mathbf{r} \to 0$, the connection component of the covariant derivative can be absorbed into the wavefunction, so that:

$$\Psi \to \Psi \exp\left(i \int_{\mathbf{r}_c}^{\mathbf{r}_c + \delta \mathbf{r}} \Gamma_i d\mathbf{r}\right) \tag{A12}$$

where $\mathbf{r}_c$ is the center of mass position and $\Gamma_i$ is the spin-connection matrix. For a Gaussian wavepacket with spread $\zeta$ and momentum $\mathbf{k}$, the wavefunction Equation (A12) takes the following form:

$$\Psi(\mathbf{r}, \mathbf{k}) = \frac{1}{\sqrt{2\pi\zeta^2}} \begin{pmatrix} 1 \\ 0 \\ 0 \\ -1 \end{pmatrix} e^{i\int \Gamma_i d\mathbf{r}} e^{-\frac{|\mathbf{r}|^2}{4\zeta^2} + i\mathbf{k}\cdot\mathbf{r}}. \tag{A13}$$

This wave-function minimally couples the standard Dirac equation to curved space through the spin connection. Defining the Berry connection as:

$$\mathcal{A}_n^i(\mathbf{R}) = i\langle\Psi(\mathbf{R})|\partial_{\mathbf{R}}|\Psi(\mathbf{R})\rangle \tag{A14}$$

for some parameter space $\mathbf{R}$ and eigen-function $n$. The Berry phase can be calculated from the complete loop integral of the connection, according to:

$$\gamma = \oint_0^{2\pi} \mathcal{A}(\mathbf{R}) g_{\mathbf{R}}^{ij} d\mathbf{R}. \tag{A15}$$

In a similar manner to the treatment of the Aharonov–Bohm effect from Berry [37], we define the slow and fast coordinates $R$ and $r$ respectively, such that $\Psi(R, r) \rightarrow \Psi(r - R)$. The wave-function takes then the form:

$$\Psi_r(R - r) = \frac{1}{\sqrt{2\pi\zeta^2}} \begin{pmatrix} 1 \\ 0 \\ 0 \\ -1 \end{pmatrix} e^{i\int \Gamma_r dr} e^{-\frac{|R-r|^2}{4\zeta^2} + ik(R-r)}. \tag{A16}$$

From Equation (A14), the explicit form of the wavefunction implies that $\mathcal{A}^i = \text{Tr}\,\Gamma_i$. The implication of this result is that the Berry connection and curvatures can be directly related to the real space affine connection and Ricci curvature tensor under suitable conditions.

As a consequence, the phase change of a wavepacket moving around a closed loop, can be calculated from the Berry phase. Integrating naively around a closed loop:

$$\gamma = \oint_0^{2\pi} \text{Tr}\langle\Psi_r|\partial_r\Psi_r\rangle g^{11} d\mathbf{r}, \tag{A17}$$

where Tr denotes the trace of the resulting matrix and takes into account the spinorial character of the Dirac wavefunction.

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
