# Peer review of "Quantized Alternate Current on Curved Graphene"

_condensedmatter, doi:10.3390/condmat4020039_

Round 1

Reviewer 1 Report

This is an excellent manuscript worthy of publication. The authors get to the salient points, make them clearly and make no attempt to obfuscate the methods employed to arrive at their results.   I do not know of any other work that has employed QRWs to curved graphene.  I am much less familiar with the computational scheme used to calculate the center of charge results, but the authors do a good job of laying out the procedure to the point where a patient reader could implement the method his or herself without undue difficulty. 

A few minor issues may be worthy of consideration:

In several sentences in the manuscript statement, there should not be a comma’s preceding the reference bracket; please bear in mind this may be a stylistic convention and may not be followed in the first language of the authors.  In any case, the ‘rule’ is not applied uniformly throughout the work. 

I would ask that the authors clarify via reference to a textbook, which convention they have chosen for their verbeins. It appears they have chosen one that appears in Kaku’s Quantum Field Theory text, but I cannot verify my memory is accurate; I am not in a location that allows me to check.

Finally, I believe this statement needs further elaboration:

“The frequency of these Bloch-like oscillations is quantized according to η. Finally, as the center103 of mass density is equivalent to a driven oscillating current, the system might be implemented as a104 periodic, quantized oscillating current device.”

It’s not immediately clear to me why the COMD is equivalent to a driven current if no voltage source is there for which a comparison could be drawn.

Author Response

a ) This is an excellent manuscript worthy of publication. The authors get to the salient

points, make them clearly and make no attempt to obfuscate the methods employed

to arrive at their results. I do not know of any other work that has employed QRWs

to curved graphene. I am much less familiar with the computational scheme used to

calculate the center of charge results, but the authors do a good job of laying out the

procedure to the point where a patient reader could implement the method his or herself

without undue difficulty.

A few minor issues may be worthy of consideration:

In several sentences in the manuscript statement, there should not be a commas preceding

the reference bracket; please bear in mind this may be a stylistic convention and may

not be followed in the first language of the authors. In any case, the rule is not applied

uniformly throughout the work.

We thank the referee for this comment. The commas have been accordingly removed

in the revised manuscript.

b ) I would ask that the authors clarify via reference to a textbook, which convention they

have chosen for their verbeins. It appears they have chosen one that appears in Kakus

Quantum Field Theory text, but I cannot verify my memory is accurate; I am not in a

location that allows me to check.

We apologize for neglecting this reference, we have added the following reference in the

manuscript:

”[35] M. Kaku,Quantum Field Theory: A Modern Introduction. Oxford University

Press, 1993.”

c ) Finally, I believe this statement needs further elaboration:

The frequency of these Bloch-like oscillations is quantized according to . Finally, as the

center103 of mass density is equivalent to a driven oscillating current, the system might

be implemented as a104 periodic, quantized oscillating current device.

Its not immediately clear to me why the COMD is equivalent to a driven current if no

voltage source is there for which a comparison could be drawn.

We thank the referee for this excellent remark. The conclusion was indeed poorly

justified. We have added the following:

”Finally, some forward moving charge,even if driven, will experience an equivalent

transverse oscillating motion, therefore the system might be implemented as a periodic,

quantized oscillating current device.”

Reviewer 2 Report

See report

Author Response

a ) REFEREES REPORT ON THE MANUSCRIPT ENTITLED: QUANTIZED ALTER-

NATE CURRENT ON CURVED GRAPHENE BY K. FLOURIS, S. SUCCI, AND H. J.

HERRMANN SUBMITTED TO CONDENSED MATTER REF. CONDENSEDMATTER-

469937

In their submitted paper, the authors propose the generation of a quantized oscillating

current on curved graphene by introducing and numerically solving a Dirac equation in

curved space. They also consider a set of semi-classical equations of motion, relating

Berry to real space curvature.They have in view applications in conjunction with trapped

fermions for the realization of quantum cellular automata. The authors claim that

The Dirac equation in curved space describes quantum relativistic Dirac particles (e.g.

electrons ) moving on arbitrary manifold trajectories. I think that such a statement is

rather audacious. What is the physical meaning of the mass term appearing in such

(respectable, of course) mathematical generalisations? The physical foundations of the

original Dirac equations for massive spin 1/2 particles derive from Poincare Wigner

symmetry requirements for Physics in flat Minkowskian space-time, for which the mass

term is the invariant rest mass of the particle. I invite the authors to support their

statement with sound explanations and physical justifications.

We thank the referee for this excellent comment. The following has been added the

manuscript to clarify and justify the comment.

”The covariant derivative ensures the independence of the Dirac equation of the coordinate basis. The covariance is satisfied by the connection coefficients which can be

interpreted physically as a vector potential. The Poincare symmetries are obeyed by

the Dirac equation ensuring the special relativistic nature of the wavefunctions. The

mass term represents the Minkowski metric invariant rest mass. Interactions add to

an effective mass by the very definition of covariant derivative, which places the vector

potential on the same mathematical basis as a physical mass. Graphene is modeled by

a mass-less Dirac Hamiltonian.”

b ) Besides this point, one can understand that the use of Dirac equation in curved space

time in the case of Graphene is justified from the similar forms of Hamiltonians (5) and

(6).

Moreover, the content is correct, well motivated, illustrated and documented. Particu-

larly interesting is the authors interpretation of their result as a geometrical analogue of

the Bloch oscillations, Just a typo: Page 1, line 15, Ahronov

The typo was corrected.
